# Comparability of the Effectiveness of Different Types of Exercise in the Treatment of Achilles Tendinopathy: A Systematic Review

**DOI:** 10.3390/healthcare11162268

**Published:** 2023-08-11

**Authors:** Aikaterini Pantelis Sivrika, Eleni Papadamou, George Kypraios, Demetris Lamnisos, George Georgoudis, Dimitrios Stasinopoulos

**Affiliations:** 1Department of Physiotherapy, University of West Attica, 28 AgiouSpyridonos Str., Egaleo, 12243 Athens, Greece; epapadamou@uniwa.gr (E.P.); gkypraios@uniwa.gr (G.K.); ggeorge@uniwa.gr (G.G.); dstasinopoulos@uniwa.gr (D.S.); 2Department of Health Sciences, European University Cyprus, 6 Diogenous Str., Engomi, Nicosia 22006, Cyprus; d.lamnisos@euc.ac.cy

**Keywords:** Achilles tendinopathy, exercise, load, rehabilitation, pilates, randomized controlled trial

## Abstract

Achilles tendinopathy (AT) is a common condition both in athletes and the general population. The purpose of this study is to highlight the most effective form of exercise in managing pain-related symptoms and functional capacity as well as in a return to life activities, ensuring the quality of life of patients with AT, and creating a protocol to be used in rehabilitation. We conducted a systematic review of the published literature in Pubmed, Scopus, Science Direct, and PEDro for Randomised Controlled Trials concerning interventions that were based exclusively on exercise and delivered in patients 18–65 years old, athletes and non-athletes. An amount of 5235 research articles generated from our search. Five met our inclusion criteria and were included in the review. Research evidence supports the effectiveness of a progressive loading eccentric exercise program based on Alfredson’s protocol, which could be modified in intensity and pace to fit the needs of each patient with AT. Future research may focus on the optimal dosage and load of exercise in eccentric training and confirm the effectiveness of other type of exercise, such as a combination of eccentric–concentric training or heavy slow resistance exercise. Pilates could be applied as an alternative, useful, and friendly tool in the rehabilitation of AT.

## 1. Introduction

Tendon injuries are common both in athletes and the general population [1] with many researchers trying to understand the mechanisms of injuries and to propose the optimal treatment in many different cases [2]. Among them, Achilles tendinopathy (AT) is one of the most common tendinopathies of the lower extremities with an estimated prevalence of 2–3 cases per 1000 patients in the general population. However prevalence rates can reach 50% in specific populations, such as athletes [3,4,5,6,7].

AT is an umbrella term used to describe clinical conditions of overuse around the Achilles tendon characterized by pain, swelling, and reduced functional capacity [8,9]. AT etiology is still unclear; however, it appears to be multifactorial. The absence of inflammatory cells does not mean that inflammatory mediators are not involved in tendinopathy while a number of theories have been developed in an attempt to explain the pathogenesis, recurrencies, chronic pain, and dysfunction that develops [10]. The tendon “progressivity” model proposed by Cook and Purdam [11] was revised in 2016 resulting in four categories of tendon pathophysiology: reactive, dysfunctional, degenerative, and reactive to degenerative tendinopathy.

The diagnosis of tendinopathy is based mainly on the patient’s history and a physical examination [12]. Magnetic resonance imaging and ultrasound are used to confirm pathology with the latter having been suggested as important in the diagnosis and the route of tendinopathy rehabilitation [13]. AT has a negative effect, both on physical activity and functional capacity, on quality of life and on productivity [14].

Conservative treatment in AT includes medical and physical therapy. Physiotherapy includes electrophysical means and manual techniques [7,15,16,17,18,19]. Exercise seems to be the intervention with the highest level of evidence [10,20,21,22]. Clinical practice guidelines report strong evidence for the use of various types of exercise, such as eccentric, concentric, isometric, and heavy slow resistance exercise [23]. Exercise provides mechanical load to the tendon in order to manage symptoms and promote strength, endurance, and satisfactory leg function [21]. However, in most trials, diverse interventions were screened while the optimal load, the duration of intervention, or the speed vary, not to mention the different outcome measures used, which all together limit the certainty of conclusions [7,21]. Historically, one of the most established exercise regimens for AT is eccentric strengthening, which as part of the rehabilitation of tendon injuries was created by Curwin and Stanish [24]. Later, Alfredson [25] modified this protocol and over time the studies evaluating its effectiveness resulted in significant benefits [26]. The effects of exercise may differ based on patient characteristics and prior treatment for AT as well as unilateral/bilateral symptoms and duration of symptoms [6] while the mechanism of the effect of exercise on the pathological process of healing is at best undefined [9]. However, it seems that the success in AT management would not only be to choose the best form of exercise, but also to be able to create personalized programs based on each individual and to empower education and activity advice [7].

The purpose of this systematic review is to study the most effective form of exercise in managing pain-related symptoms and functional capacity as well as return to life activities ensuring the quality of life of patients with AT and to create an optimal exercise protocol to be used in AT rehabilitation.

## 2. Materials and Methods

### 2.1. Search Strategy

The systematic review was conducted based on PRISMA 2020 guidelines [27]. The review protocol was registered in PROSPERO database (ID: CRD 42022365982). For data extraction, duplicates, and data selection, the management reference software Zotero Version 6.0.26 was used. Duplicates were removed automatically, yet another manually check was performed. Search was carried out independently by two researchers (A.P.S. & G.K.) in Pubmed, Scopus, Science Direct, and PEDro. Possible disagreements were resolved by a third researcher (E.P.). There was no time limit for the bibliographic search. Keywords used for the search included terms related to Achilles tendinopathy, exercise, pilates, and research design (Appendix A). A backward search was also performed on the references of the studies.

### 2.2. Inclusion–Exclusion Criteria

The design and definition of the inclusion–exclusion criteria were carried out using PICOS study design framework [28]. The participants were 18–65-year-old athletes and non-athletes diagnosed with AT without limitation in the period of time of onset of symptoms. The intervention was based exclusively on exercise. Studies were excluded if the intervention was based on other non-exercise physical therapy means and in case of combination with other non-exercise-related physical therapy means or other types of treatments (pharmaceutical treatment, infusions). Studies comparing patients with different types of AT (insertional or mid-portion) were not included. Studies were excluded for patients with a history of surgery around the ankle, repeated sprains, rupture of the Achilles tendon, ligamentous problems, or other injuries of the lower extremity. In addition, studies with patients with neurological problems–systemic diseases–metabolic diseases (e.g., diabetes) and patients receiving medication or infusions were excluded. Outcome measures concerned pain questionnaires, such as the VAS, VISA-A scales, improvement in strength, and a return to activities. Regarding the design of the study, randomized controlled studies or clinical studies in AT without time limitation were sought.

### 2.3. Methodological Quality

To assess the risk of bias of the studies, the 12 Furlan criteria were used [29] (Appendix A). Each criterion is answered by ‘yes’, ‘no’, or ‘unclear’, where ‘yes’ indicates that the criterion has been met and therefore indicates a low risk of bias. A study is classified as ‘low risk of bias’ if it meets at least six of the twelve criteria and if there are no serious losses in any group. Studies with significant losses to follow-up or those in which fewer than 6 of the criteria were met were classified as having a ‘high risk of bias’. The criteria were exhibited in Greek in a free translation by the two authors, who jointly conducted the bias control of the studies of the present systematic review.

### 2.4. Level of Research Evidence

The evaluation of the effectiveness of the interventions was carried out with the evaluation scale of the level of research evidence of Van Tulder [30] (Appendix A). This is a type of qualitative analysis, which consists of different levels of research evidence about the effectiveness of the intervention, taking into account the results and methodological quality of studies.

## 3. Results

### 3.1. Description of Studies

A total of 5235 studies were identified from the initial database search (Figure 1). Four studies met the eligibility criteria and were included in the systematic review. One study was identified from the backward search.

### 3.2. Characteristics of Studies

The characteristics of included studies are presented in Table 1. The studies have been conducted in different countries and specifically in Canada [31], Sweden [32], Korea [33], Denmark [34], and the Netherlands [35]. Studies have been conducted in a total of 191 patients, including non-professional athletes (recreational sports) [31,34,35]. Patients’ age ranged from 18 to over 65.

In three studies, patients were diagnosed with mid-portion AT [32,34,35] while in the other two studies, the type of AT is not mentioned [31,33]. Diagnosis was based in addition to a clinical examination on a soft tissue ultrasound in three studies [32,33,34]. The duration of patients’ symptoms has varied between studies ranging from 1 month [31] to 120 months [32].

### 3.3. Methodological Quality

All studies were at low risk of bias as they met more than 6 of the 12 criteria (Table 2). Regarding randomization, studies used sealed envelopes [32,35], random number tables [33], and a computer program [34]. One study did not report the method of randomization [31]. With the exception of one study [31], there was allocation concealment. The criteria of blinding of the therapist delivering the intervention and blinding of the patients during the delivery of the intervention were not met in any study. The criterion of blinding of the outcome assessor to the intervention was met in two studies [31,35]. There was no significant dropout from the program in any study, and an analysis was performed in the group to which patients were originally allocated. Bias in deriving results is unclear in all studies. In all studies, the samples were similar in terms of key prognostic indicators, and concomitant or similar interventions were avoided.

### 3.4. Interventions

In all studies, the intervention group received a progressive loading eccentric exercise program while the control group received a different exercise program (Appendix A). Specifically, in three studies, the control group received a concentric exercise program [31,32,33]; in one study, it received a combination of concentric–eccentric exercise [35]; and in one study, it received a program of heavy slow resistance training [34]. The duration of the intervention was 12 weeks in 4 of the studies [31,32,34,35] while in one study [33], it was 8 weeks. In four of the five studies, eccentric loading exercises were performed seven days per week and two times per day [32,33,34,35] while in one study [31] exercises were performed six days a week and one time a day.

In one study [32], the eccentric exercise group followed Alfredson protocol with progressive loading while the concentric exercise group followed a program that included the most common exercises used in AT. Patients were instructed to continue the program, even if they experienced pain during performance. They received practical instructions and a manual but carried out the program on their own and were checked by a physiotherapist at week six. Patients were allowed to participate in activities, such as walking or running, provided they caused mild symptoms and no pain. In another study [33], the intervention group received a combination of Curvin and Stanish and Alfredson protocols and involved progressive loading eccentric exercises. An assistant demonstrated the exercises, and patients were instructed to return to the previous week’s program if pain occurred. In addition, in this study, unlike the others, there was supervision in the execution of the exercises.

### 3.5. Compliance

Patient compliance data on the exercise program are reported in two studies [34,35]. In the first study [34], the mean rate of adherence was 78% in the intervention group and 92% in the control group, and this difference was statistically significant. In the second study [35], the mean compliance rate was 74.1% and 77.3% for the intervention and control groups, respectively, and the difference was not statistically significant.

### 3.6. Results of Outcome Measures in the Intervention and Control Groups

Pain was the outcome measure assessed in common across all studies in the systematic review. In four studies, the evaluation was done with the Visual Analogue Scale [32,33,34,35] while one study used subjective self-report by patients on a scale from 1–10 [31]. In two studies, which were performed in non-professional athletes, pain was assessed specifically in relation to daily activities and sports [34,35] while in one study, it was assessed during activity (running or walking) [32]. A second common outcome measure assessed in the two studies conducted in non-professional athletes was symptoms and functioning during sports activities with the VISA-A scale [34,35]. The other outcome measures were not common among studies. Alfredson’s protocol led to a statistically significant improvement in symptoms and functionality during daily and sports activities but not in quality of life levels [35]. Furthermore, Alfredson’s protocol resulted in statistically significant reductions in pain, statistically significant improvements in knee and calf muscle strength and endurance, and an overall balance index, dexterity, and agility [33]. An eccentric exercise program [34] led to statistically significant improvements in the VISA-A scale and pain (VAS heeland VASrunning), both after the completion of the intervention (12 weeks) and in follow-up (52 weeks). Statistically significant improvements are also reported in tendon swelling and tendon neovascularization. Finally, one study [32] found a significant reduction in pain scores in the intervention group. Similar positive results are also found in another study [31] with eccentric exercise contributing to statistically significant changes in pain reduction, maximum torque, and a return to activities (Appendix A).

### 3.7. Groups Comparison

Alfredson’s and Silbernagel’s protocols [35] were similar one year after completion of the intervention without statistically significant differences between groups. Likewise, one study [34] found no statistically significant differences between the two groups in any outcome measure and at any time point of assessment (12 weeks and 52 weeks), indicating that eccentric exercise and heavy slow resistance exercise had a similar benefit to patients. One study [31] found that an eccentric exercise protocol was equally effective in reducing pain and returning to previous activity levels while differences in mean and peak torque in dorsiflexion and plantar flexion did not have a statistically significant effect. The rest of the studies have shown that eccentric exercise compared to concentric exercise can lead to better results. In one study [33], there were statistically significant differences in the outcome measures (pain, knee extension and ankle dorsi-plantar flexion strength, plantar flexion strength, dynamic balance, dexterity, and agility) while in another study [32], patients in the eccentric exercise group showed significantly greater pain improvement and treatment satisfaction than in the concentric exercise group (Appendix A).

### 3.8. Exercise Effectiveness

Furlan’s criteria showed a low risk of bias with rates of 58–83% and low loss rates. Also, in all studies the findings were positive. The studies involved protocols with minor differences between them in terms of the choice of exercises, intensity, frequency of execution, outcome measures, and duration of the intervention. With the exception of one study [34] where one group performed heavy slow resistance training, all other studies compared the effectiveness of eccentric and concentric exercise.

Specifically, of the five studies, three compared the eccentric with the concentric [31,32,33]. The studies showed low risk of bias and low attrition rates as well as consistent positive findings in favor of eccentric exercise. From these data, strong research evidence for the choice of eccentric exercise is concluded.

One of the studies compared the eccentric with a combination of eccentric–concentric [35], resulting in positive results for both types of exercise. Since the study presented a low risk of bias and was based on the Van Tudler scale, there is moderate research evidence. The same result is obtained from another study [34] where one group performed heavy slow resistance exercise.

In conclusion, from this systematic review, it appears that there is strong research evidence for the choice of eccentric exercise and moderate for the choice of eccentric–concentric combination as well as high resistance exercises at a slow tempo.

## 4. Discussion

This systematic review studied the most effective type of exercise applied to the rehabilitation of AT in order to manage symptoms, improve functional capacity, return to daily activities, and quality of life. Since there is a lack of studies comparing different types of exercise in AT rehabilitation exclusively, it is essential to conduct studies where exercise is not combined with other therapeutic modalities or manual therapy techniques. Most studies include diverse interventions, dosage, and load progression, not to mention uncommon outcomes. For this reason, drawing a safe and consistent conclusion is limited.

Rehabilitation in AT is a major theme of clinical interest since it is a very common condition not only in athletes, but also in the general population. Historically, rest combined with pharmaceutical–analgesic–anti-inflammatory therapy and other therapeutic interventions used in rehabilitation have been replaced with functional rehabilitation, showing significantly more positive results [36,37].

Positive results have been obtained from numerous studies where exercise for Achilles tendinopathy is applied (eccentric, concentric, isometric, high-resistance exercises at a slow pace) with or without other therapeutic modalities [11,34,38,39,40,41,42] while a big discussion emerges around the effect of eccentric exercise. The most extensive protocol for the treatment of chronic Achilles tendinopathy was published by Alfredson et al. [25].

The effectiveness of this protocol In reducing the pain and symptoms related to AT is also supported in the present study, where a 12-week program seems to be sufficient for significant improvements, both in the short and long term [31,32,34,35]. Previous studies have reached similar results [43,44]. It appears, however, that compared to the effectiveness of concentric exercise, the findings do not follow a specific trend, as in some studies, eccentric and concentric exercise programs were considered equivalent in their effectiveness [35] while in others, eccentric exercise produced better results [31,32,33]. The fact that in one study [31] the results were not statistically significant probably was due to the limited sample size. Nevertheless, a careful examination of the results shows that in the eccentric group, the results were more positive in the outcome measures. In four of the five studies, the intervention concerned the same period of time (12 weeks) [31,32,34,35] while with the exception of one study [34], in all studies a similar, the exercise program was applied to intervention and control groups in terms of dosage, content, and frequency.

Any differences in results may be due to the characteristics of the study samples. For example, one study [36] was conducted in young men only while the other studies involved patients of both sexes and a wider range of age. In two studies [32,33], the duration of symptoms was longer. The details of the procedure of the intervention may also have affected the results, such as supervision of the patients’ exercise [33], in contrast to the other studies where patients performed the exercises at home with less supervision. In one study [35], the patients used other interventions, but the study was not excluded because the intervention involved a long period of time, the use of other modalities–drugs was limited in number of people and frequency. Consequently, it is considered that the results have not been affected.

A more positive outcome seems to be in favor of eccentric exercise programs. The results would possibly be even more positive if, for example, the sample was larger in number, homogeneous in characteristics, and with the appearance of symptoms in a similar period. Trying to interpret the effect of eccentric exercise in AT recovery, a considerable number of studies have used eccentric protocols versus rest, light exercise, high resistance at a slow pace, concentric, a combination of them, or another eccentric protocol. A previous study [45] reached positive results in the eccentric group, which although they were not statistically significant, were considered sufficient by the authors because apart from the positive results in a percentage of 50–60%, it is a low cost approach that is applied without or with little supervision in a very common musculoskeletal condition. An important finding in a previous study [46] was that the eccentric group reported a reduction in pain at the myotendinous junction. The behavior of the tendon in activities involving running or jumping is of interest because the fascia and muscle are shortened while the tendon is lengthened, and this possibly affects the myotendinous junction [47]. The same authors [46] reported that the reduction in symptoms was possibly due to the reduction in neovascularization, which was not confirmed in their study. In another study, researchers [20] compared groups with eccentric exercise where one of the groups was allowed to participate in activities that included running or jumping while these patients used a pain monitoring model. In this study as well as the one conducted by Stasinopoulos and Manias [41], the positive results were attributed to the way the burden (load) is applied to the tendon, the intensity determined by the patient, and the speed of execution. In a recent review, it is stated that load tolerance (type, intensity, speed of execution) is a determining factor both in the assessment and in the management of AT [7]. Based on the above, a possible interpretation could be that in widely used eccentric protocols, the exercises are performed at a low speed and the loads are increased according to the patient’s symptoms, allowing the tissue to make the appropriate adaptations.

Another possible explanation could be that the Achilles tendon is an energy-storing tendon, which reacts to forces with strain, and has a high percentage of glycosaminoglycans compared to position tendons, so it behaves and reacts differently to the lengthening–shortening cycle included in eccentric exercise [48]. In addition, in the lengthening–shortening cycle, neuromuscular adaptations are included [49].

Both eccentric and concentric protocols affect the muscle and not the tendon by making modifications in its length depending on the type of contraction while a number of theories are developed to justify how the different types of contractions affect a non-contractile structure [37,50]. The tendon is an elastic structure placed in line with the contracted part of the muscle, absorbing, storing, and releasing energy like a spring. The behavior and adaptation of the tendon to exercise depends on the type of contraction, the magnitude of the load, the range of motion, the speed, the number of repetitions, the number of sets, and the rest time between sets of the exercises. Therefore, a low-load, high-velocity, high-repetition program enhances endurance while a high-load, low-velocity, limited-repetition program is used to increase strength [37].

The maximal forces are similar to the myotendinous set in eccentric and concentric exercise [36]. This is in contradiction to the Stanish hypothesis [24]. The differences in tendon’s strength and length affect the myotendinous junction possibly like the bone where appropriate mechanical stimuli increase bone density. Thus, as the increase in the frequency of the loads provides the appropriate mechanical stimuli that lead to an increase in bone density, the corresponding changes in the loading cycles of the myotendinous set create conditions and mechanisms that justify and interpret the positive therapeutic effects of eccentric loading in AT. In practice, both peak forces and changes in tendon’s length do not show differences in eccentric and concentric contraction while it appears that fibroblasts and collagen tissue’s behavior in both forms of contraction are similar. Finally, the mechanism of the “oscillations” is caused by the eccentric exercises in addition to the stimulus that possibly leads to the reorganization of the tissue resulting in the improvement in strength and endurance, and it is very likely that they also have a positive effect on balance [36]. Likewise, in a recent systematic review [50], it is supported that both types of contractions produce similar behavior in collagen, which proves that fibroblasts are affected in a similar way, causing similar tissue changes.

It is reported that the reduction in pain may be due to an increase in the strength of the gastrocnemius muscles achieved by the application of eccentric forces, or possibly due to the fact that as the length of the tendon increases, smaller loads are exerted, or because eccentric loading causes changes in substances that affect the tissue metabolism resulting in pain reduction [32]. The increase in muscle strength is confirmed by the studies where isokinetic dynamometer was used, although possibly due to the limited sample size, it was not statistically significant [31,33]. In both studies, the measurements were performed at the same speed, which reinforces the results regarding plantar flexor strength. Another important element that needs more study is the relationship between speed and power. Although it is reported in the literature that in the eccentric contraction the force increases as the speed increases [51,52], the speed used in both studies was low (30 degrees). Given that other muscle groups have been studied in the literature, the relationship between contraction execution speed and force for the plantar flexors is expected to be further investigated in order to draw safe conclusions.

It is also worth mentioning that the decrease in the strength and endurance of the gastrocnemius muscles is associated with an increase in the predisposition to AT in runners without previous symptoms [53]. The increase in the strength and endurance of these muscles, as demonstrated in the studies of the present systematic review, reinforces the choice of the eccentric as the most effective form of exercise.

Another factor that could justify the reduction of pain with the application of exercise is the release of opioids in the brain [54]. Although the exact mechanism remains unknown, the application of eccentric exercise could justify its choice when both types of exercise are applied at the same time.

It could also be argued that eccentric exercise appears to affect strength possibly due to improvements in the abnormal morphological changes that occur in the tendon in tendinopathy. The reduction in strength is due to the difficulty caused by damage to the structure of the tendon. Changes in load-transferring, energy, and proprioceptive messages result in muscle atrophy and loss of power and disorders in factors related to balance, dexterity, and neuromuscular coordination [55]. One study [9] argues that the eccentric form of contraction may promote tissue remodeling through the reorientation of its collagen fibers, although there is no evidence to confirm the effect through histological changes. According to one study [34], high resistance exercises performed at a slow pace are equally effective reinforcing the fact that exercise improves clinical symptoms. Further studies will confirm these findings.

The ankle joint absorbs the shock of various surfaces by adjusting the position of the body. In other words, it affects balance during standing and walking. In one study [52] that implemented balance exercises in soccer players for three years, a reduction in the occurrence of AT was observed. In another study [20], although no balance outcome measures were studied, balance exercises were also implemented in the study protocol with positive results. In one study [33], the experimental group showed a large difference compared to the control group in factors related to balance. Based on the above, it could also be argued that balance is a factor that is affected in AT and that eccentric exercise shows positive results in this outcome measure as well. However, since pain reduction was not significantly different between the two groups, it could be argued that muscle strengthening and endurance are the factors that contribute to the improvement of balance, thus reinforcing the choice of eccentric exercise in AT rehabilitation programs.

Pain differentiates the position of the joints, resulting in motor dysfunction and disruption of the proprioceptive mechanism [56]. It is argued that due to the pain and the different motor patterns in walking and running, patients with AT present a disturbed neuromuscular coordination [55]. The evolution of eccentric exercise protocols with vertical and multidirectional jumps appears to show positive effects in pain and dexterity. In addition, eccentric exercise helps to increase the height of the vertical jump, which indicates an improvement in functionality. These changes are also related to patients’ satisfaction with their treatment over time [57].

However, in one study of the systematic review [35], both the eccentric and concentric groups showed positive effects and very small differences in outcome measures, although this study had limitations due to the outbreak of the pandemic. The sample size was smaller than originally estimated, which may have influenced the results. Since these protocols are performed with little or no supervision, we are unable to know how each participant progressed to the next phase in terms of intensity. In addition, the eccentric protocol is considerably more demanding and may have affected the results. Moreover, the sample in the experimental group was younger with higher expectations influencing the results. Another important element is that although an improvement of symptoms is observed in both cases, over a period of five years, mild symptoms remain in a significant percentage of patients [3].

Last but not least, clinicians should consider conducting more well established studies where heavy slow resistance programs would be an alternative [7]. In one study of this systematic review both traditional eccentric and heavy slow resistance programs yielded in short- and long-term positive results [34].

Overall, eccentric exercise is affected by many factors, but at the same time, it acts multifactorial. It is affected by factors related to the selection of exercises and load increasing, the information and instructions of execution, the compliance of the patients, and the symptom monitoring model.

Although there is no precise explanation of the mechanism underlying, mechanical loads seem to influence the healing process and cause an increase in muscle strength. The increase in strength and related factors, the effect on nerve impulses or metabolism, and the possibility of reorientation of collagen fibers and reorganization of the tissue possibly justify the positive changes in pain perception.

In this context, new exercise approaches could potentially be explored for their effect on the conservative management of AT. Pilates, for example, is a type of exercise that has begun to be investigated in recent years for its effectiveness in injury recovery [58,59]. Pilates exercises facilitate the progressive load application to the connective tissue in conditions of a closed kinetic chain [60]. However, there is limited research of Pilates exercises in tendinopathy rehabilitation [61]. The literature has supported that Pilates can nevertheless have positive effects in other conditions, such as low back pain [62,63,64] as well as in lower limb injuries and anterior cruciate ligament tears [65,66]. In this context, it can lead to improvements in pain, muscle endurance, flexibility, dynamic balance, functional capacity, and a return to activities [67]. Only one recent study was identified that investigated whether clinical Pilates may have benefits on pain and disability levels, additional to conventional physical therapy in patients with rotator cuff tendinopathy. The researchers found improvements in nocturnal pain, pain in rotation, and in disability [61].

This systematic review supports the effectiveness of an eccentric exercise program in improving pain and functional capacity in patients with AT. Given the characteristics of Alfredson’s protocol, which is the most widely used protocol in AT, a modified eccentric exercise protocol could be adopted so that it could be prescribed and applied individually with consideration to each patient’s needs as well as to available resources. The choice of exercise should be progressive in loading, frequency, endurance, and speed while it is argued that training of correct execution is essential. Loading progression could be determined on a pain basis. Activities such as running and walking and the progression of them in intensity as well as activities involving the lengthening–shortening cycle (e.g., jumping) could be determined on an individual level.

Finally, another important conclusion is that Pilates has not been evaluated in any study to date as an exercise approach to improve the symptoms of patients with AT. The researchers wish to extend the application of Pilates to the conservative management of AT in the future.

## 5. Conclusions

Application of individually tailored eccentric exercise to AT is advocated. More studies are needed to determine the dosage and way of loading across the spectrum of AT patients in eccentric exercise protocols as well as to alternatively confirm the choice of an eccentric–concentric or heavy resistance program performed at a slow pace as correspondingly effective. Pilates could be applied as an alternative and useful tool in the management of AT.

## Figures and Tables

**Figure 1 healthcare-11-02268-f001:**
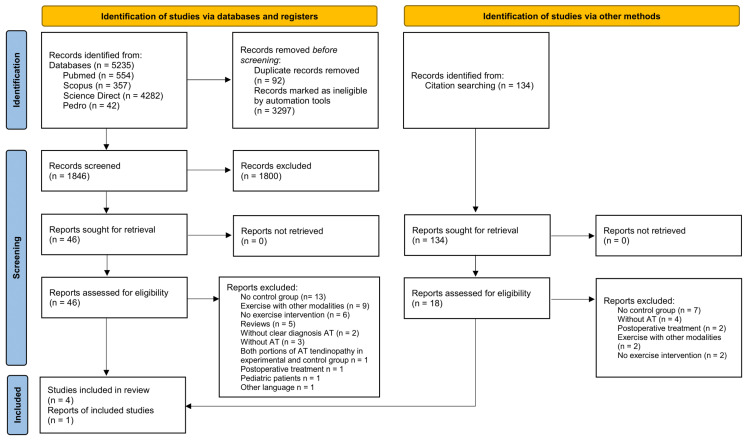
PRISMA flow diagram.

**Table 1 healthcare-11-02268-t001:** Characteristics of systematic review studies.

Authors	Sample	Sex	Symptoms Duration	Age	Intervention	Control	Outcome Measures	Results	Score
Habets et al., 2021 [35]	40 non-professional athletes Mid-portion AT AG: 18 SG: 22	Men (55%) and women (45%)	>3 months	18–65	Alfredson Group (progressive loading eccentric exercises)	Silbernagel Group (Concentric-eccentric (progressive loading)-plyometrics)	VISA-A VAS-ADL VAS-sports EQ -5D GPE 0, 12, 26 weeks, 1 year	Positive for both teams	9/12
Beyer et al., 2015 [34]	44 (out of 58 initially) non-professional athletes Mid-portion AT ECC: 25 HSR: 22	Men (68.1%) and women (31.9%)	>3 months	18–60	Alfredson Group (eccentric exercises-progressive loading)	Heavy Slow Resistance (HSR)	VISA-A VAS (heel) VAS (running) Tendon swelling (A-P, ultrasound) Neovascularization (Doppler) 0, 12, 52 weeks	Positive for both teams	8/12
Yu et al., 2012 [33]	32 EG: 16 CG: 16	Men	6 months at least	20–30	Curvin and Stanish&Alfredson et al. [24] (combination)-eccentric	Mafi et al., 2001 [32] concentric and stretches	VAS Knee and ankle muscle strength Endurance with an isokinetic dynamometer Dynamic balance (Biodex Balance Platform) Dexterity (side-step test) Agility (SargnetJump Test) 8 weeks	Pain improved more in the intervention group (IG) Knee extension, ankle dorsiflexion, plantar flexion strength (IG)-plantar flexion endurance ONLY in (IG) Dynamic balance- agility positive in both groups TBI, agility-large difference in IG	9/12
Mafi et al., 2001 [32]	44 EG: 22 CG: 22 Mid-portion AT	Men (54.5%) and women (45.5%)	3–120 months	48–58	Alfredson Group(progressive loading eccentric exercises)	Progressive loading concentric exercises	VAS Satisfaction with treatment 6, 12 weeks	Significant reduction in pain and satisfaction in IG	8/12
Niesen-Vertommen et al., 1992 [31]	17 EG: 8 CG: 9	Men (58.8%) and women (41.2%)	1 month–2.6 years	22–49	Curvin and Stanish (progressive loading eccentric exercises)	Progressive loading concentric exercises	Pain 0–10 Return to activities 0–10 Maximum torque in concentric and eccentric Plantar flexion at speeds of 30 and 50 degrees/sec Isokinetic dynamometer 0, 4, 8, 12 weeks	Pain reduction, maximum torque- return to activities positive in both groups- emphasis in the IG	7/12

**Table 2 healthcare-11-02268-t002:** Risk of bias scores of the randomized controlled trials of the systematic review ((+)—YES, (−)—NO, (?)—UNCLEAR).

Study	1	2	3	4	5	6	7	8	9	10	11	12	Maximum Score	Score	Rate
Niesen-Vertommen et al. [31]	?	?	−	−	+	+	+	?	+	+	+	+	12	7/12	58%
Mafi et al. [32]	+	+	−	−	−	+	+	?	+	+	+	+	12	8/12	67%
Yu et al. [33]	+	+	−	−	−	+	+	+	+	+	+	+	12	9/12	75%
Beyer et al. [34]	+	+	−	−	?	+	+	?	+	+	+	+	12	8/12	67%
Habets et al. [35]	+	+	−	−	+	+	+	+	+	+	+	+	12	10/12	83%

## Data Availability

All data generated or analysed during this study are included in the published article.

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
