# Peer review of "Comparability of the Effectiveness of Different Types of Exercise in the Treatment of Achilles Tendinopathy: A Systematic Review"

_healthcare, 2023, doi:10.3390/healthcare11162268_

Round 1
Reviewer 1 Report
Thank you for letting me read your manuscript. The topic is indeed interesting, even though I have some concerns regarding the need for it.
When reviewing your work, I followed AMSTAR 2 (a critical appraisal tool for systematic reviews that include randomized or non-randomized studies of healthcare interventions, or both). Reference: https://www.bmj.com/content/358/bmj.j4008
The 16 questions according to AMSTAR 2 are as follows:
-
Did the research questions and inclusion criteria for the review include the components of PICO? P, I, and O are described. But what is exercise compared to (C)?
-
Did the report of the review contain an explicit statement that the review methods were established prior to the conduct of the review, and did the report justify any significant deviations from the protocol? No pre-registration is reported (Prospero would have been expected, especially when reporting that you have followed PRISMA: http://www.prisma-statement.org/Protocols/Registration?AspxAutoDetectCookieSupport=1). What conclusions can be made if we do not know about your plan before conducting the study? This is a major limitation that needs to be discussed. Additionally, there is no data synthesis plan, which is missing.
-
Did the review authors explain their selection of the study designs for inclusion in the review? YES
-
Did the review authors use a comprehensive literature search strategy? Partial yes. It is unclear if the authors searched the reference lists/bibliographies of included studies, searched trial/study registries, or included/consulted content experts in the field.
-
Did the review authors perform study selection in duplicate? Study selection and data extraction in study duplicates are unclear. The PRISMA flow diagram is unclear, and the duplicate process should also be commented on in order to explain how study selection and data extraction were handled.
-
Did the review authors perform data extraction in duplicate? See above.
-
Did the review authors provide a list of excluded studies and justify the exclusions? No
-
Did the review authors describe the included studies in adequate detail? Partial yes.
Please always explain the abbreviations used in table S4. Doses could be relevant to include (intervention and comparator). What is the comparator? For example, Beyer et al. write: “Therefore, we sought to investigate the effect of a 12-week HSR regimen compared with the traditional eccentric loading regimen in patients with midportion Achilles tendinopathy in a randomized controlled trial with a 52-week follow-up.” – They have HSR as the intervention and ECC as the comparator. This is also a problem when looking at the results. Line 147 states, “In all studies, the intervention group received a progressive loading eccentric exercise program, while the control group received a different exercise program” – this is hereby untrue. Additionally, in table S4, “Characteristics of interventions in the intervention group” is confusing – isn't it the treatment in both the intervention and the control groups?
-
Did the review authors use a satisfactory technique for assessing the risk of bias (RoB) in individual studies that were included in the review? Yes
-
Did the review authors report on the sources of funding for the studies included in the review? No
-
Not relevant (since the present study is not a meta-analysis).
-
Not relevant (since the present study is not a meta-analysis).
-
Did the review authors account for RoB in individual studies when interpreting/discussing the results of the review? Yes
-
Did the review authors provide a satisfactory explanation for, and discussion of, any heterogeneity observed in the results of the review? No
-
If they performed quantitative synthesis, did the review authors carry out an adequate investigation of publication bias (small study bias) and discuss its likely impact on the results of the review? No
-
Did the review authors report any potential sources of conflict of interest, including any funding they received for conducting the review? No.
Results from the AMSTAR checklist: out of 14 relevant items, 3 partial yes, 8 no, 3 yes.
RESULTS The results are unstructured and are not presented clearly. Preferably, present one outcome at a time (in the text). The main result tables should not be in the supplementary file but in the main manuscript! Line 195: “The rest of the studies have shown that eccentric exercise compared to concentric exercise can lead to better results.” References are missing. Exercise effectiveness: I doubt your conclusion since it is still unclear to me what you are comparing exercise with. Or is it eccentric exercise with other training? Furthermore, you have an issue with the intervention group/control group in the article from Beyer et al., as mentioned previously. As a part of your purpose, you write that you want to “…create a protocol to be used in rehabilitation.” I do not read anything about this in neither the methods nor the results section. You discuss exercise in the discussion section, but you never add new results/knowledge in the discussions section that hasn’t been presented before.
Discussion Difficult to follow since the number of references is incorrect! Please start the discussion section with 2 lines telling the reader – what was your most interesting finding. Line 226-236 is not needed, more as a background. You should discuss your findings in relation to present literature. When I continue to read the discussion, it is far too long. Line 260-273 – is it relevant to write since these studies (ref 45-46) compare eccentric loading with other treatment modalities? Furthermore, your discussion does not mainly focus on the results from your study. It needs a major revision where the most interesting results are being discussed, and there is less discussion about areas not related to your results and aim. Were there no limitations with your study?
Finally, I wonder if this review is needed. In line 240, “Previous studies have reached similar results [43,44].” Why is your review needed? What new knowledge does it contribute? A final question/recommendation for your next manuscript – make sure to use the latest version of guidelines. You refer to PRISMA 2009 – they have been revised since then! Due to the small amount of “yes” answers following AMSTAR, together with my own judgment of this manuscript, I unfortunately need to recommend it to be rejected. My recommendations for the authors: make sure to have supervision next time, ensuring that you are following the latest guidelines and make a pre-registration of your trial. Best of luck! A last remark – your English language was well-written!"
Author Response
"Please see the attachement"

Reviewer 2 Report
The current study aimed to identify the most effective form of exercise in managing 19 pain-related symptoms and functional capacity as well as return to life activities, ensuring the quality of life of patients with Achilles Tendinopathy (AT) and to create a protocol to be used in rehabilitation. Although this issue is interesting, important points are addressed to the Authors to clarify the purpose as well as the quality of this paper.
Based on the textual content, there is a clear disconnection between Title, Introduction, main objective, and the Results/Discussion. While the Title is related to effectiveness of exercise (in general) to treat AT, the Introduction is poor in terms of information and background to support the main objective, which was destined to search for the most effective (or the best) form of exercise training intervention in AT conditions.
It is noteworthy that the Results section accentuated these divergences and did not allow us to make consistent comparisons among different exercise training interventions, as eccentric exercises constituted a common intervention used in the found studies. Consequently, the Discussion became heavy, cumbersome, and excessively speculative. A deep review of the text and structure is necessary to improve the manuscript.
Other major review questions are following:
In accordance with the Material and Methods, recruitments incorporating 20-60 years-old participants constituted an inclusion criteria for the selection of studies. However, two articles containing 18-65 and 18-60 years-old participants, respectively, were considered as included studies in this review. From the age criteria, both references must not be included.
Another criterion was “Studies comparing patients with different types of AT were not included”. However, as duration of patients’ symptoms varied from 1 to 120 months, it is noteworthy that different clinical conditions were mixed within same context. It is necessary to clarify this point from citations by searching for these information.
Based on the Conclusion, the authors argued that additional should be developed to make clear potential effects from eccentric protocols in relation to AT condition. Indeed, physiological adaptations due to exercise training are dependent on specific stress characteristics. As a consequence, comparing different interventions such as eccentric protocols versus Pilates are unusual; firstly, it is necessary to report what are the effects of interest to treat AT clinical conditions.
Author Response
"Please see the attachement"

Reviewer 3 Report
I thank the authors for the opportunity to read the work.
In my opinion, the introduction is not sufficient and does not justify revisiting the topic.
Why are Lamnisos and Georgoudis mentioned in the acknowledgments if they are the authors of the work?
The quality of figure 1 is unsatisfactory.
Material and methods described in detail. Results presented in a clear and transparent way. The discussion was conducted well.
Importance of the topic and scientific significance 3/5.
Author Response
Introduction: We will work on that.
Lamnisos and Georgoudis are co-authors. We will correct it.
Please provide us with some details for the figure which you commented "unsatisfactory" so that we can improve .
Round 2
Reviewer 2 Report
The Authors made considerations relative to the first reviewing process and, as a result, the paper was improved and it is acceptable to be published.
Author Response
Many thanks